# A Systematic Review of Machine-Learning Solutions in Anaerobic Digestion

**DOI:** 10.3390/bioengineering10121410

**Published:** 2023-12-11

**Authors:** Harvey Rutland, Jiseon You, Haixia Liu, Larry Bull, Darren Reynolds

**Affiliations:** 1School of Computer Science, Electrical and Electronic Engineering, and Engineering Maths, University of Bristol, Bristol BS8 1QU, UK; 2School of Engineering, University of the West of England, Bristol BS16 1QY, UK; jiseon.you@uwe.ac.uk; 3School of Computing and Creative Technologies, University of the West of England, Bristol BS16 1QY, UK; haixia.liu@uwe.ac.uk (H.L.); larry.bull@uwe.ac.uk (L.B.); 4School of Applied Sciences, University of the West of England, Bristol BS16 1QY, UK; darren.reynolds@uwe.ac.uk

**Keywords:** machine learning, deep learning, anaerobic digestion

## Abstract

The use of machine learning (ML) in anaerobic digestion (AD) is growing in popularity and improves the interpretation of complex system parameters for better operation and optimisation. This systematic literature review aims to explore how ML is currently employed in AD, with particular attention to the challenges of implementation and the benefits of integrating ML techniques. While both lab and industry-scale datasets have been used for model training, challenges arise from varied system designs and the different monitoring equipment used. Traditional machine-learning techniques, predominantly artificial neural networks (ANN), are the most commonly used but face difficulties in scalability and interpretability. Specifically, models trained on lab-scale data often struggle to generalize to full-scale, real-world operations due to the complexity and variability in bacterial communities and system operations. In practical scenarios, machine learning can be employed in real-time operations for predictive modelling, ensuring system stability is maintained, resulting in improved efficiency of both biogas production and waste treatment processes. Through reviewing the ML techniques employed in wider applied domains, potential future research opportunities in addressing these challenges have been identified.

## 1. Introduction

Anaerobic digestion (AD) is a biological process where microorganisms break down biodegradable material in the absence of oxygen, resulting in biogas production—a mixture of methane, carbon dioxide, and trace gases. This biogas serves as a renewable energy source, and the digestate is a nutrient-rich fertilizer. The complexity of AD, influenced by numerous variables such as substrate composition, temperature, pH levels, hydraulic retention time, and microbial community dynamics, poses challenges in monitoring and optimisation. Machine Learning (ML) has emerged as a pivotal tool in interpreting the nonlinear relationships inherent in these AD systems, enhancing control, operational safety, and performance forecasting [1,2]. Literature reviews in the domain of ML applications for AD, such as those by Cruz et al. [3] and Gupta et al. [4], acknowledge the nascent stage of ML-based solutions in AD. They focus on algorithm suitability, black-box challenges, data limitations, and potential applications such as process optimisation and kinetic parameter learning. These reviews also discuss integration challenges such as data demand and model selection issues.

This paper extends these discussions by offering a comprehensive review of ML applications in AD. We compare these applications with those in other fields to identify opportunities for cross-domain knowledge transfer. Our review addresses key research questions related to dataset characteristics, preprocessing stages, model selection, and optimisation strategies in AD contexts [3,4]. By identifying systematic gaps in the current literature, this paper aims to enhance the understanding and application of ML in AD, contributing to the advancement of this field.

### Research Questions

This paper presents a systematic review of ML applications in AD. We focus on identifying current ML trends, methods, and their impact on AD while examining the challenges and benefits of this integration. Special attention is given to future directions, including explainable ML and its role in assisting AD control. Distinct from broader reviews, this study delves into practical implementation aspects and highlights unexplored areas in ML for AD, as outlined in Table 1.

## 2. Materials and Methods

### 2.1. Research Design

The systematic literature review conducted in this study followed the procedure illustrated in Figure 1. Papers were screened using a keyword search related to the given topics, as listed in Table 2. The initial search looked at journals from Science Direct detailing the application of ML with data from AD systems. The papers gathered were then screened based on predefined inclusion and exclusion criteria (highlighted in Table 3), using both abstract and full-text reviews. Data from selected papers were then extracted and synthesized to address the listed research questions shown in Table 1. Papers that demonstrated the use of ML in a different relevant applied field were kept to enable an understanding of how ML is being used in other applied applications and to determine potential future research gaps in the AD field.

### 2.2. Metadata Extraction

Table 4 details the information that was recorded from each paper meeting the inclusion criteria. Each paper was assessed concerning the data extraction points listed DEP). Some of the points included are for purposes of documentation; reasoning is referenced in relation to the context of the related research question. To understand the pre-processing requirements for datasets used in AD experiments, data were collected from the following extraction points: DEP05 (Research field), DEP06 (Application Domain), DEP07 (Data Source), DEP10 (Dataset variable structure), and DEP08 (Experimental Scale). This was important in order to understand the context of the AD operation.

## 3. Results

### 3.1. Dataset and Data Pre-Processing

Papers meeting the specified inclusion criteria have been listed in Table 5. Studies on ML applications in AD have been conducted on various experimental scales, ranging from lab-scale experiments to industrial applications. Lab-scale experiments explore the fundamental principles of the AD process or technology and determine the feasibility before scaling up to larger-scale systems. Industrial studies, on the other hand, involve the deployment of AD processes and/or technology at a commercial scale. The large volume of data collected provides practical insights into how ML can be applied directly to real-world AD applications. As a result, the scale of the AD operation can determine whether the dataset was collected for the purpose of an ML study or provided by an external AD plant to facilitate ML investigations. Table 6 gives an overview of the datasets used by the papers reviewed in this study.

It was noted from the reviewed literature that data from multiple variables collected in AD experiments such as COD and TS are monitored offline, which can be time-consuming and expensive. As a consequence, the rate of data collection from offline variables is significantly lower than that of online parameters, such as gas composition and flow rate [14,22]. One challenge highlighted is acquiring all variable data at the same temporal intervals or taking complete measurements without missing any temporal-specific measurements. Some variables can be measured quickly using in situ sensors and meters, while others require time-consuming laboratory-based experiments. Datasets can be incomplete due to issues such as equipment failure and measurement errors. This can result in non-aligning data points in the training data, which is unsuitable for model training [10,14]. An important stage is the inclusion or exclusion of outlier values; this can be common due to sensor error (for example, an error reading may be negative or excessively out of the expected range). Once removed, methods such as listwise deletion and other forms of imputation can impede model accuracy if data trends are not properly captured.

Furthermore, datasets from biogas plants may encompass operational data unrelated to the target prediction variable [23]. Discerning parameter correlations allows for their exclusion in the data curation process, utilizing prior knowledge of the target wastewater and variable relationships. In [8], researchers first eliminated fields unconnected to biogas production attributes and removed variables indirectly linked to biogas (supplementation for missing values was unnecessary in this case).

### 3.2. ML Techniques

A wide range of models were employed in the reviewed literature; some papers conducted a detailed comparison between the performance of multiple ML models to determine which approach yields the most accurate results, whereas others conducted a more focused development of hybrid models (combining the strengths of multiple ML techniques). This section looks to focus on DEP11 (ML algorithms used) and DEP12 (Optimal models identified) to understand the pipeline required to use experimental/industrial AD to develop ML-assisted functionality.

#### 3.2.1. Traditional Machine-Learning Methods

Figure 2a presents the ML models referenced in the reviewed studies, with neural networks (NNs) being the most frequently mentioned. It is important to note that several studies focused specifically on NNs and did not compare them to alternative ML model types. Approximately 30% of the studies conducted comparative investigations of different ML techniques for anaerobic digestion (AD) applications. Other literature supported the decision to investigate and develop hybrid models by referencing comparative studies [5].

Support Vector Machines (SVMs), Random Forest, Gradient Boosting, and Decision Trees were among the other popular ML models identified. These models have been widely used in various ML-AD applications due to their ability to handle data with high dimensionality, capture complex relationships, and provide accurate predictions. Furthermore, these models can offer robust predictions, even with relatively small sample sizes [17].

##### ANN

Table 7 presents ANN applications in various domains, prominently in biogas predictions. These examples not only demonstrate the varied applications of ANNs but also underscore the importance of network structure in achieving high model performance. The careful tuning of network architecture is a critical factor in the success of ANNs as it directly influences their learning capacity and generalization ability [24].

Ref. [13] details a feedforward ANN with 3 input, 14 hidden, and 2 output neurons. Input parameters—temperature, pH, and recirculation ratio—were selected via ANOVA from a broader feature set. The model, trained on a 2-year time-series dataset of pH and temperature, underwent hyperparameter optimisation in MATLAB, employing Levenberg–Marquardt for its fitting efficacy. Comparative analysis of Tansig, Purelin, and Logsig activation functions indicated optimum performance, with Logsig in the hidden layer and Purelin in the output layer, yielding an R^2^ value of 0.9762. The study acknowledges, however, ANNs’ lower prediction efficiency compared to tree-based models.

Additionally, ref. [19] describes the application of an ANN in a UASB reactor treating SEOW for biogas production prediction. The model, structured with specific neurons in multiple layers, was developed to process variables such as influent chemical oxygen demand and pH. Over 141 days, the ANN model achieved an R^2^ value of 0.975. While this study does not detail specific activation functions or training algorithms as in [13], it demonstrates the ANN’s high predictive accuracy.

##### SVM

SVM, and its regression variant SVR (Support Vector Regression), were frequently mentioned in the reviewed literature. They were identified in [18] as the most robust models for use in developing a VFA (Volatile Fatty Acids) soft sensor. The study assessed the models’ ability to predict VFA levels, particularly in cases of system faults such as pH sensor drift. A grid search was conducted on the SVM, using a radial basis function, to optimize the cost and gamma hyperparameters, resulting in values of 245.88 and 0.0030, respectively. Although the ELM (Extreme Learning Machine) and ENN (Evolving Neural Network) models showed better results in the testing and validation phases, the SVM model proved to be the most robust for soft sensor applications. The robustness calculation is detailed under Equations (Equation 1) and (Equation 2).
(1)$$S_{i}=\frac{\sum_{k=1}^{N}{\left({\mathrm{VFA}}_{\mathrm{pred}}^{\mathrm{fa}}-{\mathrm{VFA}}_{\mathrm{real}}^{\mathrm{no}}\right)}^{2}}{N}$$
where:$S_{i}$ represents the robustness with respect to fault *i*;${\mathrm{VFA}}_{\mathrm{pred}}^{\mathrm{fa}}$ is the VFA predicted by soft-sensors during the faulty event;${\mathrm{VFA}}_{\mathrm{real}}^{\mathrm{no}}$ is the real VFA in the normal operation of the process;*N* is the number of all faulty samples.
(2)$$S=\sum_{i=1}^{m}S_{i}$$

##### Tree Models

RF, Gradient Boosting, and DT are all tree-based models, which work by splitting data into subsets based on given feature values. The final output is obtained by aggregating the predictions made at the last node [25]. These models operate by constructing a series of decision trees during training. In RF, multiple decision trees are built and trained on different subsets of the data. Each tree gives a prediction, and the final output is the aggregation (typically the mode or mean) of these predictions [26]. Gradient Boosting builds trees sequentially, where each new tree aims to correct the errors made by the previous ones [27]. DT, on the other hand, involves creating a single decision tree with a set of binary rules for decision-making, typically based on information gain or Gini impurity [28]. These models have been commonly referenced in the reviewed literature as they are well suited to non-linear relationships [29]. Additionally, the results can be interpreted, which is useful for understanding the biological reasoning behind the predictions [28]. A comparison of RF and XGBoost (among other models) is conducted in [22]. An overview of the hyperparameters used to tune the model complexity is provided. RF focuses on the number of features used for splits, while XGBoost manages tree depth and structure. Both models achieve similar Root Mean Square Error (RMSE) in the prediction of $\left(L-CH_{4}/g-COD\right)$. RF had a slightly better RMSE of 0.034 when using pH as a single training parameter, with XGBoost returning a similar RMSE of 0.035. Alternatively, when training with multiple input parameters both models achieved an RMSE of 0.032. In addition to this, ref. [8] lists the application of tree models in biogas predictions, where 10-fold cross-validation was used to ensure accurate model validation. Model parameters were selected using grid search. Compared to other ML models such as SVM and LSTM, tree-based methods yielded lower prediction errors. RF produced the lowest error, with a Mean Absolute Error (MAE) of 269 m^3^ (volume biogas measured at industrial plant scale).

The distribution of ML techniques over the reviewed studies is depicted in Table 8. The table presents a categorization of the 18 research papers based on the ML techniques employed, including traditional ML, deep learning, and traditional methods with novel modifications (Studies P4 and P18 are listed twice as they fall under both traditional ML and deep learning).

#### 3.2.2. Deep-Learning Methods

Deep learning has emerged as a promising approach in the field of anaerobic digestion. Of the reviewed papers, four studies listed the use of deep-learning variations, detailing the model setup and optimisation. The parameters from the studies are listed in Table 9. Similar to the ANN techniques discussed in this paper, the input neurons in DNNs represent variables relevant to the AD process. Optimisation methods such as PCA have been employed for this application, akin to their use in traditional ML applications. Additionally, other hyperparameters in DNNs, such as learning rate, batch size, optimizer, and dropout rate, are explored in papers detailing DNN development. These studies emphasize the application of methods such as Genetic Algorithms (GAs), grid search, and the Bayesian optimisation Algorithm (BOA) for hyperparameter optimisation.

Grid search is a computationally exhaustive method that ensures a thorough search over a given set of hyperparameter combinations [30]. Alternatively, the Bayesian optimisation Algorithm (BOA) is a more intelligent and efficient method of hyperparameter optimisation. Unlike grid search, BOA uses past evaluation results to form a probabilistic model mapping hyperparameters to the model’s performance. This approach allows for the prediction of optimal hyperparameters by balancing exploration (searching new regions of the hyperparameter space) and exploitation (focusing on regions returning promising performance) [31]. The application of BOA is detailed in [14], where the number of iterations for optimized hyperparameters was set to 50 for deep-learning models. Similarly, hyperparameter optimisation through GAs can be a more efficient tuning technique compared to methods such as grid search, making it suitable for tuning DNNs with complex search spaces. The application of tuning a DNN with a GA for the purpose of training weights and biases is detailed in [9]. The optimisation procedure performed by the GA was completed when the average relative change in the optimum fitness function attained over 100 generations was ≤$1\times 10^{-10}$, the fitness attained the value of ≤$1\times 10^{-5}$, or the generation was seen to increase. These optimisation methods can be used in conjunction with cross-validation to ensure a robust assessment of the given parameter combination.

Hybrid deep-learning architectures, which incorporate a Variational Sequence-to-Sequence (VSN) or attention mechanism, have been shown to be effective in training RNN-based models on raw datasets without prior pre-processing for missing value imputation or outlier elimination [14]. The Variational Seq2Seq mechanism enables models to better manage the intricacies and correlations within datasets and can be trained on raw data that has not been pre-processed for missing value imputation or outlier removal [32].

DNNs have proven effective in determining the relationship between water properties and biogas composition. Ref. [9] details the use of a dataset gathered from laboratory experimentation. The model identified optimal slurry ranges with high accuracy. The dataset used in this study was collected from a 24-day lab-scale BMP experiment. The input data consisted of 30 sets of slurry characteristic measurements taken before and after the experiment (each measurement consisting of 14 lab-monitored offline AD parameters). The target biogas composition data were taken as an average of three measurements recorded over the course of the BMP. To compensate for the small dataset size, the input data were randomly broadcast into six datasets. In a comparable context of wastewater analysis, a DNN-LSTM hybrid model was developed to predict NO_2_ concentrations based on wastewater properties [10]. Historical data lags of NO_2_ were utilized for step-ahead predictions in downstream time series analytics. Both slurry and wastewater analyses demonstrate the potential of deep-learning techniques in addressing complex problems within the domain of AD.

#### 3.2.3. Time-Series Implementation

ML has been applied in various areas of AD, including water analysis, methane yield prediction, system modelling, biogas yield prediction, biogas optimisation, and VFA prediction. Sixty percent of the reviewed papers describe the use of ML to make time-series predictions of methane/biogas production. This application domain uses regression models to predict the future performance of the system based on feedstock/operational characteristics. Similar time-series forecasting was implemented for VFA prediction in [5,15], providing an operator with early fault detection signals. If the VFA becomes too high the system will become unstable and stop producing biogas; use of fault detection can inform the operator to take preventative action.

The utilization of step-ahead techniques for time-series prediction tasks have been highlighted in [10] for biogas production and [22] for wastewater analysis. These approaches exhibited relatively high performance for downstream prediction tasks; a diagram outlining the training structure is shown in Figure 3. Implementing these forecasting methods showed that the accuracy of the model predictions decreased when the forecast horizon exceeded one day. Ref. [17] provides an explanation for this, attributing it to the rapid fluctuations in the feedstock. Consequently, downstream prediction may be less reliable when the model training process does not account for the forecasted feedstock change.

In the application domain of system modelling, ML models were used to provide insights into the AD process, support concept design, and aid system operations. Two papers conducted an analysis on whether or not AD can be sustainably implemented to provide bioenergy in a circular economy. Ref. [12] provides a method of waste stream viability assessment to estimate energy at scale-up using a model trained on a dataset collected from lab-scale testing. Alternatively, Ref. [15] assesses energy recovery, using decision tree classification to assist human-in-the-loop decision-making in AD control operations. The system was modelled using ML to generate desirable operational parameters and inform the control of the AD system. Ref. [20] outlines the use of RSM to optimise an ANN to predict AD conditions (reactor feed, recirculation, and temperature) optimal for methane yield. In a similar context, Ref. [21] reports on the optimisation of an MEC-AD system. An RSM was used to predict batch biogas production under three operating voltages. Particle Swarm Optimisation (PSO) was subsequently used to identify the optimal voltage that would result in maximum net energy production.

### 3.3. Feature Assessment

Anaerobic digestion involves a complex interplay of multiple factors, such as temperature, pH, feedstock composition, and microbial activity, which can influence the performance and stability of the system. The data generated from such systems can have high dimensionality, noise, and non-linearity, which makes it challenging to model using theoretical/mathematical approaches when the system possesses a higher complexity [14]. Different feature reduction techniques have been used in several studies to enhance the performance of ML models in the listed AD applications, with the most common being Principle Component Analysis (PCA) and Analysis of Variance (ANOVA).

Figure 2b indicates the common variables used for model training. Figure 4 shows how these variables relate to the overall AD operation. Papers which outline the use of ML to predict energy production from the AD system use information surrounding biogas generation as the target variable. In the context of this table, both biogas production and methane production have been listed under the category of biogas production. As mentioned in Section 1, biogas produced in AD consists of a mixture of different gas compounds. Among these compounds, methane is the predominant gas used for energy production due to its flammable properties. Methane production is thus a key component used to determine energy production in anaerobic digestion systems. Methane production is calculated by measuring the gas composition and total gas production to infer the volume of the individual gas component. In a healthy anaerobic digestion system, 55–75% of the biogas produced is methane [33]. Consequently, methane production typically follows a similar trend to biogas production. In the context of ML, these two variables will likely exhibit high collinearity. Therefore, they have been grouped together for ease of reference in this discussion.

A predicted drop in biogas can indicate that adjustment to parameters such as feed metrics, temperature, or re-circulation is required in situations such as if the system is showing signs of instability (resulting in energy drop) [34]. pH was the common variable across the reviewed studies. Multiple studies which reviewed feature importance indicated pH to have the highest feature importance when predicting biogas production. Model performance when only using pH as the data input results in lower system complexity and training time but additionally caused there to be a greater error in prediction results [22]. Reducing demand for the number of features required for model training can be beneficial for AD applications where collecting/processing samples for lab processing can be time-consuming and expensive.

### 3.4. ML Techniques That Are Not Used in AD Applications but Can Be Found in Other Applied ML Fields

A wider review of ML literature detailing non-AD related applications was conducted to assess potential future developments in intelligent AD systems. This section outlines the identified topics and gives a brief description of how the methods are currently being utilised in alternative applications.

#### 3.4.1. Reinforcement Learning Application

Reinforcement learning (RL) algorithms are a branch of ML that learns through trial and error, finding optimum rules/policies in the process [35]. The typical RL process involves generating data by interacting with the environment, evaluating the agent’s performance based on this data, and using it to improve the agent’s policy. The generated data contains information about the actions taken, the state of the environment, and the resultant reward. This approach imparts an exploratory nature to the system, where the agent’s past experiences inform future decision-making and aid in preventing over-fitting. The agent, utilizing a combination of parametric features and thresholds, receives an aggregate score reflecting the effectiveness of its actions. This score guides the agent in adjusting its strategies to minimize errors and maximize forecast accuracy.

To give an example of the application of RL, ref. [36] details the use of RL for crop yield predictions using externally sourced data spanning over a 35-year period to train a Deep Recurrent Q-Network (DRQN). This dataset included a wide range of features, such as temperature, precipitation, soil pH, and nutrient content, among others. The RL approach is compared to other ML methods, such as Deep LSTM, ANN, Gradient Boosting, and Random Forest. Evaluation metrics such as R^2^ and MAE are used to compare model performance. The performance results indicate the superior performance of the DRQN compared to other ML methods, achieving an R^2^ of 0.87 and an MAE of 0.13, reflecting a low average error in yield predictions. Alternatively, ref. [37] investigates the application of RL for online energy management in smart grids for the purpose of optimizing energy consumption schedules in buildings. The approach employs two RL algorithms, Deep Q-learning and Deep Policy Gradient, both adapted for simultaneous multiple actions. The effectiveness of these methods is validated using the Pecan Street Inc. database, which includes data on photovoltaic power generation, electric vehicles, and building appliances. The study demonstrates the ability of the proposed RL methods to adaptively and efficiently manage energy consumption and costs in real-time, outperforming traditional methods in terms of peak reduction and cost minimization.

In the context of an AD system, applying RL can be particularly advantageous for optimizing multiple objectives such as feeding, heating, and mixing [38]. These elements are critical for the efficiency and effectiveness of AD processes. Multi-objective reinforcement learning is a method for balancing control of multiple parameters and optimizing these concurrent objectives within an AD system [39]. By integrating RL into AD systems, it is possible to enhance decision-making processes, leading to more effective management and operation of these systems. Similar methods may be implemented in AD-related applications to account for differences in operational parameters from lab-based experiments to scale-up applications, allowing the systems to calibrate online and adjust to inevitable parameter differences such as feedstock type and temperature. Initial investigation into this work has been conducted in [40], where the approach is tested in simulation. This study demonstrates the feasibility of RL for optimizing methane production in a simulated anaerobic digestion system. However, barriers to real-world implementation are acknowledged, such as a need for accurate modelling of built systems and accounting for real-world feedback and parameters such as sensor noise and lag.

#### 3.4.2. ML/IoT Practises

ML and IoT technologies are starting to have a significant impact on the farming industry as this enables the monitoring of climate factors, soil characteristics, and soil moisture to improve crop production [41]. Precision agriculture is a similar concept that uses sensing and analysis tools to improve crop yields, reduce labour time, and effectively manage fertilizers and irrigation processes. Precision agriculture employs data from multiple sources to improve crop management strategies. Similar frameworks to what is proposed can be applied to AD scenarios in ML tools and implemented at an industrial level. The use of these tools is highlighted to assist analysis and decision-making. Applications using big data and data mining practises allows patterns to be identified which would not be observable on a smaller scale.

#### 3.4.3. Explainable AI

AI-based systems can be used to automatically recognize patterns in data and assist subject matter experts in their evaluations, particularly in situations where complex knowledge and strategies are involved [42]. However, for these AI-based systems to be successfully integrated into industrial processes such as AD, they must be trustworthy and comprehensible to decision-makers, who can provide analytical reasoning and explain the decisions made by a system. This requires an advanced management process to develop trust in the actions, inference mechanisms, and results of the AI-based systems. An example of this is shown in [43], where the ensemble of 5 ML models is used to predict the distribution areas of seagrasses. The study introduces the explainable AI (XAI) method to provide reasoning through systematic decomposition. The methodological framework reveals the internal operation patterns of the model. This can provide explanations to assess the effects of changes in the training variables (environmental factors) and provide an interpretation of the prediction results given. Understanding the causes of the resulting impact on seagrass allows for target measures to be given to promote sustainable conservation and restoration. This can improve the understanding of AD system control where information provided by the XAI enables relationships to be inferred, which would be difficult to interpret through traditional analysis methods.

## 4. Discussion

It was highlighted in Figure 2a that Neural Networks (NN) were the most frequently mentioned ML models for AD applications. However, other models referenced in the reviewed literature, such as Support Vector Machines (SVMs), Random Forest, Gradient Boosting, and Decision Trees, were also popular. These models were widely used due to their ability to handle data with high dimensionality and provide accurate predictions, even with relatively small sample sizes. It should be considered that multiple papers outline ML-AD solutions by only detailing the development of NN solutions without providing a comparison to other traditional ML techniques. The prevalence in this research field may be due to their ability to effectively handle intricate relationships in dynamic data, making them well-suited to model the interactions present in complex systems. Their success and popularity in a variety of application domains further supports their widespread adoption and effectiveness across diverse application contexts [44]. However, it should be noted that the papers reported in this systematic review focused more on the development of ML-AD technologies in the research domain. If these prediction and forecasting tools were to be implemented at an industrial level, it would be important to interpret how decisions are being generated. In this context, traditional techniques such as Decision Trees, XGBoost, and Random Forest might be considered more feasible for implementation at an industrial scale. The technique of implementation would also need to be considered when assessing model viability with integration in AD operation at an industrial scale. As implementation of an NN can be more computationally demanding and may limit continuous retraining, this poses significant challenges for real-time, resource-constrained industrial applications [45].

The most common variables identified in model training include temperature, pH, biogas production, Chemical Oxygen Demand (COD), and reactor feedrate. Among these factors, pH stands out as the most prevalent variable across multiple studies. While pH is often recognized for its high feature importance in predicting biogas production, emphasizing its significance in the modelling process, the relationship between pH and biogas yields is complex and sometimes ambiguous. This complexity stems from the pH’s dependence on the buffering capacity of the medium in the bioreactor [46]. Buffering capacity, which refers to the medium’s ability to resist pH changes, depends on various factors, including the composition of the feed and operational conditions. This complexity presents challenges in accurately measuring and incorporating these nuances into ML models [47].

In terms of AD operation, pH is regarded as a significant variable for AD predictions as it directly relates to system stability and impacts microbial activity essential for biogas production [48]. However, the efficacy of ML models in this context is potentially limited by the difficulty in capturing the multifaceted interactions influenced by pH, including the buffering capacity [49].

Furthermore, pH can have a direct relationship with other variables such as feedrate, and subsequently, the COD of the feed. If the feedrate at which the organic matter is fed to the system is too high, this can lead to an accumulation of VFA, causing a drop in pH. This, in turn, can inhibit activity from the microorganisms responsible for methane production, thereby directly impacting biogas yield [50]. On the other hand, COD is a measure of organic matter in the feed and indicates the potential biogas that can be produced [38]. Feeding high COD to a digester can lead to VFA production (similar to the effects of a high feedrate) [51]. As such, with the vast majority of ML-AD applications relating to the prediction of methane/biogas production, it can be inferred why both parameters play a crucial role in making such predictions.

The challenge in finding representative experimental data for training such formal models further complicates the development of robust ML applications in AD. Future research should focus on methods to better measure and incorporate factors such as buffering capacity and feed composition into ML models, as well as on improving data collection and representation to more accurately reflect the dynamics of AD systems.

### 4.1. What Are the Identified Benefits of Implementing ML Techniques to Assist AD Operation?

In the reviewed studies, several highlighted benefits of applying ML in AD were repeated in multiple papers spanning the different specified application domains. One of the most prominent benefits is the reduced hardware complexity and costs associated with using ML as a ‘soft sensor’ [18,52]. The use of ML algorithms to predict system behaviour offers the potential to require less regular lab analysis and reduce demand for human labour. The fault detection systems outlined in this section can enable operational issues to be prematurely addressed prior to the system becoming unstable.

Another benefit prominently highlighted across the reviewed literature is the enhancement in waste treatment modelling and control. ML algorithms aid in predicting gas yield, which is crucial for managing the treatment and energy demands of the system efficiently. Moreover, ML algorithms fortify system performance through the continuous identification of operational anomalies, enabling early interventions to mitigate potential system failures, which in turn can diminish operational costs. Additionally, by forecasting gas yield, ML assists in assessing treatment demands and providing insight for operational adjustment, thereby facilitating efficient biogas production. The perpetual monitoring driven by ML can reduce the demand for manual oversight, which can enable more efficient system operation and facilitate the implementation of more automation within the systems.

### 4.2. What Challenges Are Faced in Applying ML to Improve the Autonomy of AD Applications?

Several common trends were identified relating to the limitations of using ML in AD applications. In relation to DEP15, a major limitation of this technology highlighted in the literature is the lack of sufficient training data. ML algorithms require large datasets to train on, and in many cases there may not be enough data available to effectively train the model [53]. A common reason highlighted in the literature is difficulties in acquiring all variable data at the same time-step intervals. Another challenge is the difficulty in comparing different AD systems due to variations in operating conditions, making it challenging to develop a universal model for multiple applications.

Table 6 highlights that the majority of lab-based experiments are conducted in batch mode rather than using continuous flow. Operating under such conditions can limit opportunities for automating a continuous system. Without validation on continuous flow systems, it is difficult to ascertain how ML technologies can be implemented to automate aspects of control in industrial AD plants.

As mentioned, a lack of sufficient training data was a commonly highlighted issue when approaching the development of ML-AD solutions [5,13]. An issue not extensively covered in the literature is how models trained on an AD system may be re-implemented on a different AD system, a process known as transfer learning [54]. Models trained on one anaerobic digestion system may not be directly transferable to another digester if the operational parameters do not align with the original model [20]. In addition to this, differences in bacterial community composition in the systems will need to be accounted for in the process of initial calibration [55]. Monitoring systems in biogas plants will need to be assessed to ensure infrastructure can facilitate the data demand required for useful predictions [17]. This is important when considering whether or not ML technologies can feasibly be implemented in AD operations at an industrial scale.

### 4.3. ML Contributions to AD Practices

Assessing DEP07 and DEP08 showed that a significant proportion of the reviewed literature discusses and analyses data retrieved from industrial biogas facilities as opposed to lab-based operations. The most common research methodologies focused on the use of data from offline waste analysis variables (such as COD) and operation data (feed volumes) to predict the biogas production from the AD system. DEP10 indicated that a wide variety of models were used for this application domain, including traditional and deep-learning techniques. Overall, this showed ML to be well-suited to processing the data used in AD applications due to its ability to extract features from datasets of high dimensionality, enabling it to predict the energy production expected from the AD. If implemented into a live AD operation, this would enable the operator to adjust how the system is controlled accordingly to prevent systems from entering an unstable state.

Of the reviewed literature, DEP11 indicated that a significant proportion of the papers reviewed had selected a model based on literature/case studies, rather than directly comparing model performances with ranging complexities. This was often in the context of developing and tuning deep-learning models or assessing how deep-learning models can be further optimised through ensemble methods. If ML is to be used in a live AD system, consideration may be necessary regarding how the model will be retrained and adapted, as such computational demand will need to be compared to infer industrial feasibility [56].

## 5. Conclusions

This review reveals the challenges and potential of applying ML to AD, particularly the scarcity of training data and the variability in AD systems. These issues hinder the development of universal models, requiring system-specific approaches and adjustments for bacterial community variations. Choosing models from the literature rather than performance comparison calls for more rigorous evaluation for industrial use. The use of established deterministic models to expand datasets in cases of scarcity or gaps due to system downtime can offer a promising solution to these challenges [57]. Integrating such models can alleviate data limitations and boost model robustness, underscoring the need for thorough comparative analysis and evaluation to ensure ML’s industrial applicability in AD.

Addressing these challenges requires investments in robust data acquisition frameworks, closer academia–industry collaborations for faster validation of ML models in continuous systems, and a more stringent evaluation of ML models to ensure their industrial applicability. By tackling these focal points, future research can facilitate the effective integration of ML into AD operations, steering the AD field towards enhanced efficiency and automation.

## Figures and Tables

**Figure 1 bioengineering-10-01410-f001:**
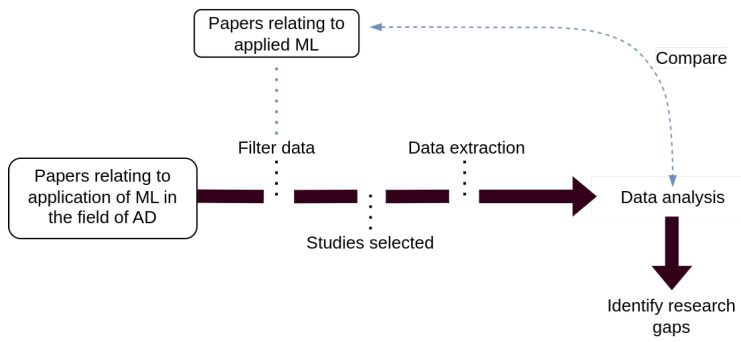
Research process diagram.

**Figure 2 bioengineering-10-01410-f002:**
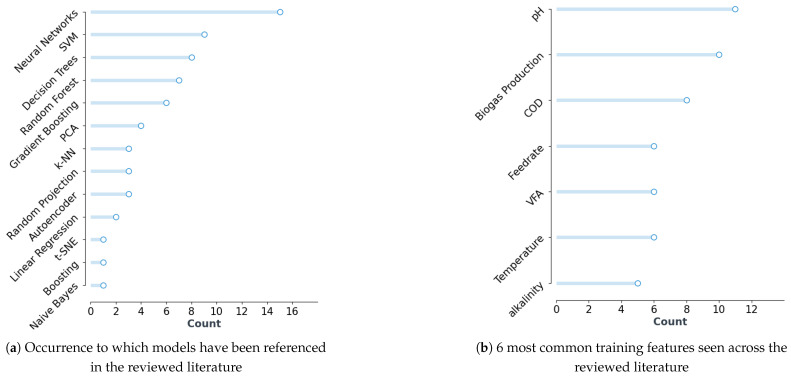
(**a**) Details the top-referenced ML models in the literature, highlighting the number of papers citing each model. (**b**) Shows key variables commonly found across the ML-AD training datasets, highlighting the number of papers citing each variable.

**Figure 3 bioengineering-10-01410-f003:**
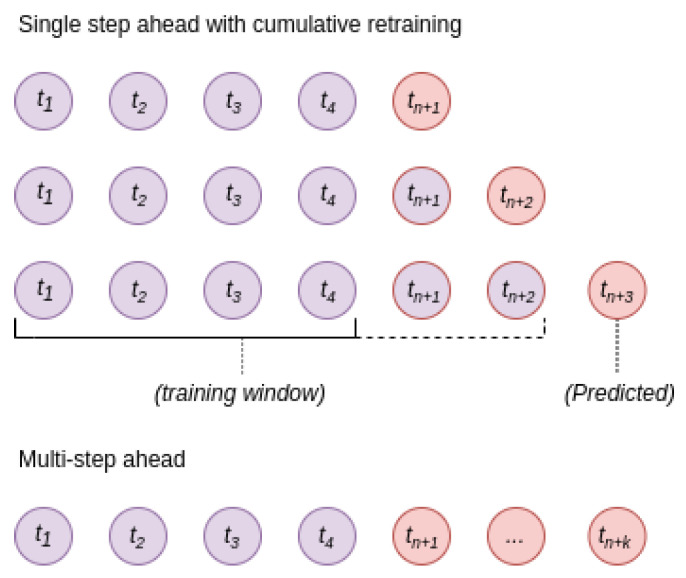
This figure illustrates two forecasting methods in anaerobic digestion from the literature. ‘Single-step ahead forecasting’ predicts one future point at a time, using an expanding training dataset. ‘Multi-step ahead forecasting’ predicts several future points simultaneously.

**Figure 4 bioengineering-10-01410-f004:**
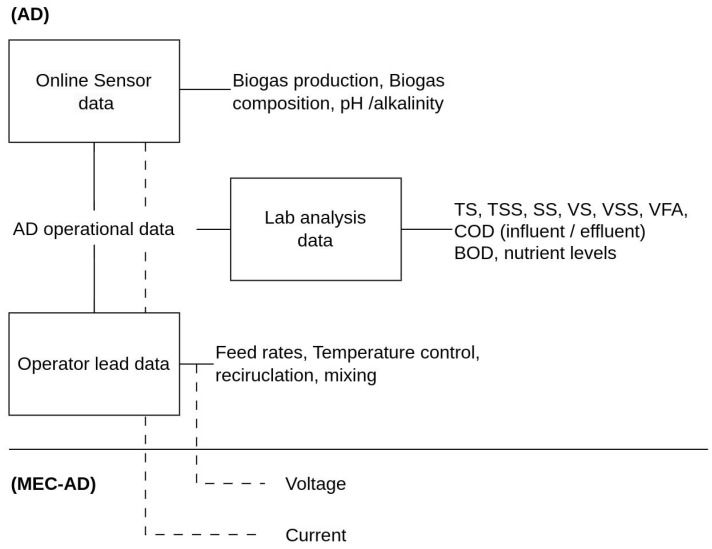
Diagram illustrating AD dataset variables identified in the reviewed literature.

**Table 1 bioengineering-10-01410-t001:** Research Questions (RQs) and Rationales.

Reference	Research Question	Rationale
RQ1	How is ML being practically used in the field of AD?	This question aims to detail particular applications where the components of an AD system have been used to train ML models and outlines the role these models play in AD operation.
RQ2	What ML techniques have been selected in the reviewed literature?	This research question was selected to understand how researchers have selected and compared different ML techniques and if this relates to the application domain and data structure.
RQ3	What are the identified benefits of implementing ML techniques to assist with AD operation?	This aims to provide an up-to-date understanding of the advantages of particular ML techniques have over traditional statistical approaches.
RQ4	What challenges are faced in applying ML to improve the autonomy of AD applications and systems?	This question looks at the challenges/considerations which should be addressed in order to remove the need for human-in-the-loop operation.
RQ5	What ML techniques have been successful in comparable process applications?	The rationale behind this research question is to identify potential ML techniques that have been successful in other fields but have not yet been applied in AD.

**Table 2 bioengineering-10-01410-t002:** Keyword search terminology.

Sub-Category	Keyword
Machine learning	‘machine learning’, ‘learning’, ‘performance’, ‘prediction’, ‘intelligence’
AD/MEC-AD/Waste Water	‘gas’, ‘methane’, ‘anaerobic’, ‘digestion’, ‘waste’, ‘treatment’

**Table 3 bioengineering-10-01410-t003:** Inclusion/Exclusion Criteria.

Criteria Type	Point	Rationale
Inclusion 1	Date of publication is after 2018	This criterion ensured that the studies considered were recent and focuses on the most up-to-date techniques and technologies in ML and sensing.
Inclusion 2	Poses an application of ML in comparable research applications	The studies being considered were relevant to the research question and use ML in comparable contexts.
Exclusion 1	Does not present primary research or is a review paper	This review considered primary research studies rather than reviews to ensure the depth of analysis required to answer the research question was met.
Exclusion 2	Duplicate of a previously evaluated document	Meta-analysis conducted was based on unique work and not incorrectly skewed by duplicated studies.
Exclusion 3	Paper does not show the application of ML algorithms with subject-relevant datasets	The studies being considered should be based on empirical data rather than pure theory.

**Table 4 bioengineering-10-01410-t004:** Data Extraction Point (DEP) collection form, with their descriptions and corresponding research contexts, linked to Research Questions RQ1–RQ4 (from Table 1) to inform the response to RQ5.

Data Extraction Point	Field	Description	Research Context
DEP01	Study Number	The number assigned to the included papers for reference	For documentation purposes, to facilitate easy identification and reference to the individual studies within this review
DEP02	Paper Title	Title of publication	For documentation purposes to provide a brief overview of the study’s focus and scope
DEP03	Publication Year	The year in which the paper was published	To allow chronological development of the research topic to be traced over time
DEP04	Journal	The publication venue	To identify which outlets have predominantly published in this field
DEP05	Research Field	Context of work (waste water, anaerobic digestion, MEC-AD)	RQ1
DEP06	Application Domain	The role played by ML	RQ1
DEP07	Data Source and Description	Where the dataset was obtained: Extracted from literature, provided by industrial scale operation, data collected for study purpose	RQ2
DEP08	Experimental Scale	The size of AD operation from which data were collected	RQ1
DEP09	Performance Optimisation	How the ML optimised the overall AD operation	RQ1
DEP10	Variable Structure	Variables referenced in study dataset	RQ2
DEP11	Machine-Learning Algorithm(s)	What machine-learning algorithms have been applied to an AD context	RQ2
DEP12	Listed Optimal Model	The best-performing model identified in the given study domain	RQ2
DEP13	Feature Engineering Methods	Pre-processing methods applied to dataset for viable ML training	RQ2
DEP14	Highlighted Benefits	Benefits that ML has brought to the specified AD operation	RQ3
DEP15	Challenges/Limitations	Highlighted challenges and limitations encountered when implementing ML solutions in the AD application domain	RQ4

**Table 5 bioengineering-10-01410-t005:** List of papers included in the review.

Paper Number	Title	Year	Journal
P1	Brewery wastewater treatment plant key-component estimation using moving-window recurrent neural networks [5]	2020	IFAC
P2	Modelling and simulation of co-digestion performance with artificial neural network for prediction of methane production from tea factory waste with co-substrate of spent tea waste [6]	2021	Fuel
P3	Estimation of in situ biogas upgrading in microbial electrolysis cells via direct electron transfer [7]	2021	Bioresource Technology
P4	Key waste selection and prediction improvement for biogas production through hybrid machine-learning methods [8]	2022	Sustainable Energy Technologies and Assessments
P5	Integrated deep-learning neural network and desirability analysis in biogas plants [9]	2020	Energy
P6	Ingredient analysis of biological wastewater using hybrid multi-stream deep-learning framework [10]	2022	Computers and Chemical Engineering
P7	Modelling biogas production from anaerobic wastewater treatment plants using radial basis function networks and differential evolution [11]	2021	Computers and Chemical Engineering
P8	Constructing a smart framework for supplying the biogas energy in green buildings using an integration of response surface methodology, artificial intelligence, and petri-net modelling [12]	2021	Energy Conversion and Management
P9	Process modelling and optimisation of methane yield from palm oil mill effluent using response surface methodology and artificial neural network [13]	2023	Journal of Water Process Engineering
P10	Prediction of biogas production in anaerobic co-digestion of organic wastes using deep-learning models [14]	2021	Water Research
P11	Integration of swine manure anaerobic digestion and digestate nutrients removal/recovery under a circular economy concept [15]	2021	Journal of Environmental Management
P12	Plant-scale biogas production prediction based on multiple hybrid machine-learning technique [16]	2022	Bioresource Technology
P13	Exploring available input variables for machine-learning models to predict biogas production in industrial-scale biogas plants treating food waste [17]	2022	Journal of Cleaner Production
P14	Data-driven techniques for fault detection in anaerobic digestion process [18]	2020	Process Safety and Environmental Protection
P15	Use of artificial neural network and adaptive neuro-fuzzy inference system for prediction of biogas production from spearmint essential oil wastewater treatment in up-flow anaerobic sludge blanket reactor [19]	2021	Fuel
P16	Optimisation and performance evaluation of response surface methodology(RSM), artificial neural network (ANN), and adaptive neuro-fuzzy inference system (ANFIS) in the prediction of biogas production from palm oil mill effluent (POME) [20]	2022	Energy
P17	Artificial intelligence-based modelling and optimisation of microbial electrolysis cell-assisted anaerobic digestion fed with alkaline-pretreated waste-activated sludge [21]	2022	Biochemical Engineering Journal
P18	Retraining prior state performances of anaerobic digestion improves prediction accuracy of methane yield in various machine-learning models [22]	2021	Applied Energy

**Table 6 bioengineering-10-01410-t006:** Overview of datasets used for ML model training and development in AD Studies. The table summarizes the source of the database and the duration of the data collection period.

Paper Number	Data Source	Dataset Length
P1	Industrial-scale AD facility	19-day (10-day training, 9-day cross-validation)
P2	Lab data from AD experiment with tea factory waste	49-day bmp test
P3	Two datasets used from previous research	71-day and 138-day datasets, operating under a continuously fed artificial waste stream
P4	Open source data from Industrial-scale AD facility	1340-day
P5	Lab experiment	24-day bmp test running co-digestion of food and animal waste
P6	Industrial wastewater plant	5-day (10second interval with live sensors)
P7	Data provided by industrial paper mill	389-day (75%, 25% training cross-validation)
P8	AD operating with palm oil mill effluent	Not explicitly stated
P10	Data provided by full-scale municipal wastewater treatment plant	731-day and 27-day (used to demonstrate the use of DL for AD process optimisation)
P11	Concept modelled on SISTRATES^®^ waste management system	-
P12	Industrial food-waste treatment plant	-
P13	Industrial food-waste treatment plant	360-day
P14	Data provided using simulated dataset using benchmark simulation model from international water association	-
P15	Lab experiment treating synthetic spearmint essential oil wastewater with continuously fed reactor	141-day
P16	Industrial scale palm oil mill effluent AD facility	-
P17	Data extracted from lab-based study reported in the literature	20-day batch experiment
P18	Lab-based AD reactor (working volume 18L), operating under varied COD-based OLRs using food waste	630 day

**Table 7 bioengineering-10-01410-t007:** Overview of the models assessed in the reviewed studies, highlighting the optimal model and corresponding AD application context of ML implementation.

Paper Number	Models Listed	Optimal Model/Hybrid Optimisation	Application Domain	KPI Metrics
P1	RBF-ANN	Application of moving window (MV-RBF-ANN)	Hybrid model development of an ANN to provide time-series prediction of parameters such as VFA and biogas composition.	-
P2	ANN	No comparison listed	Determining co-digestion ratio of 2 waste streams to maximise methane yield.	R^2^
P3	ANN, Linear regression, Tree, SVM, Ensemble, Gaussian Process Regression	Hybrid ANN (NARX-BP ANN)	Predicting methane production from biogas upgrading in MECs via direct electron transfer (DET)	RMSE, R^2^, MSE, MAE
P4	RF, XGBoost, DT, SVM, Linear regression, LSTM, KNN	Hybrid RF+LSTM	Analysing the key components in wastes streams to improve biogas generation predictions	MSE, RMSE, MAE
P5	DNN	No comparison listed	Prediction of biogas response to slurry properties	R^2^, AR^2^, MSR
P6	DNN, LSTM, CNN	DNN-LSTM hybrid	Analysis of waste stream ingredients to inform purification controls	RMSE, CORR
P7	SVM, GP, linear/quadratic regression, CART, MLP, RBF-ANN	RBF-ANN (trained with systematic fuzzy means) hybrid	Modelling biogas production in a wastewater plant using industrial-scale data	MAE, R^2^
P8	RT, RF, ANN, ANFIS	ANFIS	Forecasting accumulated biogas production	CORR, MAE, RMSE, RAE, RRSE
P9	BDD-RSM, ANN	ANN	Modelling methane and hydrogen sulfide production	RSME, R^2^
P10	LSTM variations	DA–LSTM–VSN	Forecasting downstream methane production	R^2^, NRMSE, MAE(%)
P11	DT	No comparison given	Implementation of classification to assist human-in-the-loop decision-making in AD control operations.	-
P12	ELM hybrid variations using SMOTER and GA	SMOTER-GA-ELM	Forecasting biogas production	R^2^, MAE
P13	XGBoost, RF ANN, SVM	RF	Forecasting up to 10 days of downstream biogas production using data concerning feedstock properties	R^2^, RMSE
P14	ENN, ELM, SVM	SVM	Predicting total VFA for use in a system fault detection framework	NRMSE, R^2^
P15	ANN, ANFIS	BP-ANN	Modelling biogas production using data from a lab-scale UASB reactor	R^2^, RMSE, RRMSE(%)
P16	RSM, ANFIS, ANN	ANFIS	Prediction of biogas production/methane yield and optimisation of controllable parameters to maximise gas production/methane yield.	R^2^, MSE, MAE
P17	RSM, ANN, PSO	No comparison given	Use of an ANN (tuned using RSM) to generate biogas predictions. PSO was used to optimise the potential difference applied to the MEC electrodes to maximise the energy output	MSE
P18	RNN, XGBoost, RF, SVR	RNN	1-step ahead with retraining method to predict AD biogas production using data detailing prior state in the time-series data	RMSE

**Table 8 bioengineering-10-01410-t008:** Categorization of the reviewed studies by ML Technique.

Evaluated ML Methods	Paper Number
Traditional ML	P2, P3, P4, P7, P8, P11, P13, P14, P15, P16, P17, P18
Deep Learning	P4, P5, P6, P10, P18
Traditional ML with NovelModification	P1, P9, P12

**Table 9 bioengineering-10-01410-t009:** Overview of NN configurations and optimisation parameters in DL studies on AD. Studies which did not clearly detail the listed information were not included in the table.

Paper Number	DNN Points of Interest	Output Context	Loss Metric	Hyperparamter Optimisation
P4	RF-LSTM Input layer = 10 neurons (key operation variables) Training parameters = learning rate (0.01), epochs (50), batch size (32), adam optimizer, drop out rate (0.5)	Biogas production	MSE	Grid search Cross-Validation
P5	DNN -LSTM Input layer = 14 neurons (representing slurry properties) Layer of neural network fully connected	Biogas compound (%) prediction	MSE	GA
P10	DA-LSTM-VSN (no data preprocessing required) 17 neurons in the input layer	Gas prediction	MSE	BOA
P18	LSTM (single-step ahead and multi-step ahead comparison) Comparison of 1 and 4 neurons in the input layer, comparing predictions using pH to a model trained using pH, Alkalinity, COD removal, VFA concentration Hyper-parameters included in LSTM optimisation (final values not included) = learning rate, number of hidden nodes, batch size	Methane Yield	RMSE	10-fold cross-validation

## Data Availability

All data underlying the findings of this review are unrestricted and fully available. The data that support the findings of this study are contained within the article and its referenced literature.

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
