# Peer review of "A Systematic Review of Machine-Learning Solutions in Anaerobic Digestion"

_bioengineering, 2023, doi:10.3390/bioengineering10121410_

Round 1

Reviewer 1 Report

Comments and Suggestions for Authors

 Some remarkes and recommendations:

1. The structure and overall presentation of the article make it suitable for a Machine learning journal, but not for a biologically focused journal. A typical example is the inclusion in the article of paragraph 3.4. To make the article suitable for this journal, it should be emphasized what the results are related to the process itself in the cited articles, and not what algorithms are used (paragraph 4. is not enough, because there are many general conclusions without specific and in-depth examples).

2. The cited literature is new and up-to-date (only 3 out of 43 cited are older than 5 years - about 7%); But only 41 references. (2 are repeated) for overview are few. The cited 3 reviews from 2022 include more references. Even in the first cited review, the references are more, although it concerns a narrower field of AD. At least twice moor references are needed.

3. There is no comparison of the proposed algorithms with the much more developed ones based on the numerous known AD models. Particularly attractive is a possible combination of these two types of algorithms - for example, the use of known mathematical models for training the proposed algorithms (as done in [Simeonov I., E. Chorukova, Neural networks modeling of two biotechnological processes, Proc. Second IEEE International Conference on Intelligent Systems, June 2004, pp.331-336]).

4. There are repeated references: 3=5; 4=6.

5. The observation that pH is a significant variable is true, but not only for the reasons stated. The dependence of microbial activity and hence biogas yields on pH is very complex and in many cases ambiguous. E.g. the pH value strongly depends on the so-called "buffering capacity" of the medium in the bioreactor, which in turn depends on many factors and is difficult to measure. Even more difficult would be to find representative experimental data with a view to training such a formal model. Therefore, I believe that such an approach "a priori" has limited capacity.

6. The conclusion is too general and shallow in the specific area.

Conclusion

My personal opinion is that the proposed algorithms have a future in the modeling of AD processes, but a deeper knowledge of these processes or joint work with specialists in the field and those who have obtained significant achievements in "classical" modeling is needed.

Author Response

Thank you for this feedback. We recognize that the original version of the article may have focused more on the algorithms than on their application to the biological processes involved in AD. The revisions aim to rectify this by offering a more balanced view, where the technical details of ML techniques are directly related to their practical implications and benefits in biological systems. Below, you will find our responses to the additional questions raised in your feedback.

The structure and overall presentation of the article make it suitable for a Machine learning journal, but not for a biologically focused journal. A typical example is the inclusion in the article of paragraph 3.4. To make the article suitable for this journal, it should be emphasized what the results are related to the process itself in the cited articles, and not what algorithms are used (paragraph 4. is not enough, because there are many general conclusions without specific and in-depth examples).

In section 3.4 the discussion of RL applications in non-anaerobic digestion (AD) fields has been expanded, providing specific examples and quantitative results. This includes detailed analysis of RL in crop yield prediction and smart grid energy management, supported by performance metrics like R2 and MAE. These examples demonstrate the practical implications and effectiveness of RL in the relevant fields.

Additionally the revision of this section aims to provide an idea of how RL can contribute to the understanding and optimization of AD. We have provided an example of how RL can be utilized to optimize multiple objectives within AD systems, particularly in a simulated environment, which offers insights of the future potential this can bring to AD scenarios and research.

The cited literature is new and up-to-date (only 3 out of 43 cited are older than 5 years - about 7%); But only 41 references. (2 are repeated) for overview are few. The cited 3 reviews from 2022 include more references. Even in the first cited review, the references are more, although it concerns a narrower field of AD. At least twice moor references are needed.

This point has been taken into consideration. In agreement with your feedback, we have included more seminal references in our revised manuscript to support key points in the discussion, alongside recent studies. We hope this helps to strengthen the arguments presented and ensure a comprehensive overview of the subjects presented.

There are repeated references: 3=5; 4=6.

Our apologises for overlooking the errors, the references have been updated and checked in the revised version.

There is no comparison of the proposed algorithms with the much more developed ones based on the numerous known AD models. Particularly attractive is a possible combination of these two types of algorithms - for example, the use of known mathematical models for training the proposed algorithms (as done in [Simeonov I., E. Chorukova, Neural networks modeling of two biotechnological processes, Proc. Second IEEE International Conference on Intelligent Systems, June 2004, pp.331-336]).

Thank you for your suggestion. While our initial screening did not identify papers specifically addressing this context, we recognize the importance of integrating established AD models with ML developments. In response, we have updated our manuscript's conclusion to emphasize the need for rigorous comparative analysis between ML algorithms and established AD models. This revision underscores the potential of using these models to augment datasets in scenarios of data scarcity.

The observation that pH is a significant variable is true, but not only for the reasons stated. The dependence of microbial activity and hence biogas yields on pH is very complex and in many cases ambiguous. E.g. the pH value strongly depends on the so-called "buffering capacity" of the medium in the bioreactor, which in turn depends on many factors and is difficult to measure. Even more difficult would be to find representative experimental data with a view to training such a formal model. Therefore, I believe that such an approach "a priori" has limited capacity.

In our studies, we noted the importance of pH in trained ML models. This observation primarily relates to dataset examination and dimensional reduction in ML. However, recognising the focus of this journal, we have updated our discussion in accordance with your suggestion to more thoroughly explore the role of pH in AD. The revised section delves into the complexities surrounding pH, particularly its dependency on buffering capacity and the associated challenges in ML modelling. Additionally, we examine how pH interacts with critical variables like feedrate and COD in the context of AD operations, and discuss the consequential impacts on biogas production.

The conclusion is too general and shallow in the specific area.

The conclusion has been updated in an atempt to make it more direct in summarising the results of the paper and highlighting the main findings. We highlight how use of ML in AD require system-specific approaches, while emphasizing the importance of comparative model performance analysis rather than literature-based model selection.

Reviewer 2 Report

Comments and Suggestions for Authors

ABSTRACT

Line 2. rephrase "and offers significant potential to interpret the intricate parameters needed for system operation and optimisation."

line 6. repharse "notably in terms of monitoring equipment employed".

line 7. this statement can be removed from the abstract "Such variations contribute to data scarcity, a challenge further exacerbated by the inherently complex bacterial community within AD systems and the diverse operational procedures implemented."

INTRODUCTION

Line 23. please name those multiple variables

Line 28-29. in references 3 and 4, please add the surname of authors, then the number of reference

line 31. rephrase "Both sources concur"

Line 48. "Challenges and limitations" can be removed.

maybe section 1.1 can be reduced and fused with the main introduction body?

section 1.2 tends to repeat the last paragraph of the introduction

MATERIALS AND METHODS

Table 3 is named before table 2

line 86. please name the science directories used

please improve the quality of figure 1

please add the keywords used to refine the search

in table 4, what does it means RQ1-4?

Table 5 is a result rather than a method

RESULTS

remove lines 104-105

section 3.1 is more a method than a result

table 9 is too far from the first mention in the text

figure 2 appears first than their first mention in the text

if ANN is so heavily used, why there are not "several" references?

What SVR means?

what VFA sensor means?

there is no methods added to create the "tree method"

there is no explanation on figure 3

please improve quality of figure 4

CONCLUSIONS

Please simplify them, its really long and is not easy to read the main findings 

Author Response

Thank you for the detailed feedback clearly outlining the necessary changes. We have implemented all the specific changes requested. Below, you will find our responses to the additional questions raised in your feedback.

ABSTRACT

Line 2. rephrase "and offers significant potential to interpret the intricate parameters needed for system operation and optimisation."

This change has been actioned.

line 6. repharse "notably in terms of monitoring equipment employed".

This change has been actioned.

line 7. this statement can be removed from the abstract "Such variations contribute to data scarcity, a challenge further exacerbated by the inherently complex bacterial community within AD systems and the diverse operational procedures implemented."

This change has been actioned.

INTRODUCTION

Line 23. please name those multiple variables

Varibles have been named, ‘The complexity of AD, influenced by numerous variables such as substrate composition, temperature, pH levels, hydraulic retention time, and microbial community dynamics, poses challenges in monitoring and optimization.’

Line 28-29. in references 3 and 4, please add the surname of authors, then the number of reference

This change has been actioned.

line 31. rephrase "Both sources concur"

This change has been actioned.

Line 48. "Challenges and limitations" can be removed.

This change has been actioned.

Maybe section 1.1 can be reduced and fused with the main introduction body?

Thank you for this suggestion. The introduction has been revised to incorporate what was originally in section 1.1 into the introduction. Offering a concise of the contribution this paper aims to provide.

Section 1.2 tends to repeat the last paragraph of the introduction

In response to this feedback, we have revised the paper to better distinguish the Introduction, which now provides a succinct context, from Section 1.2, focused solely on detailed research questions. This aims to remove the overlap between the 2 sections.

MATERIALS AND METHODS

Table 3 is named before table 2

This change has been actioned.

line 86. please name the science directories used

The science directory used has now been named in the text.

Please improve the quality of figure 1

This figure has been updated.

Please add the keywords used to refine the search

This information has been listed in table 2.

In table 4, what does it mean by RQ1-4?

Thank you for highlighting this, the acronymn refers to research questions 1 to 4. However, I've adjusted the table description in an attempt to make this clearer and have adjusted table 1 description to list the acronym.

Table 5 is a result rather than a method

This table has been moved to the result section.

RESULTS

remove lines 104-105

This change has been actioned.

section 3.1 is more a method than a result

This change has been actioned.

table 9 is too far from the first mention in the text

The layout of the paper has now been adjusted to better align figures and tables to the reference point in the text.

figure 2 appears first than their first mention in the text

This change has been actioned.

If ANN is so heavily used, why there are not "several" references?

Thank you for highlighting this important aspect. In response to your concerns about the initial lack of extensive references, the revised manuscript now enriches Section 3.2.1. Traditional Machine Learning Methods with an additional, detailed example that more effectively demonstrates the application of Artificial Neural Networks (ANNs). Additionally, Table 8 lists further papers that explore ANNs in comparison to other models.

Furthermore, the manuscript delves deeper into deep learning, a specialized subset of neural networks characterized by their layered architectures, in Section 3.2.2 Deep Learning Methods. This section aims to offer a broader perspective on the diverse applications of neural networks.

What SVR means?

Support vector regression, the meaning of this acronym has now been listed on line 147. and highlighted in the abbreviations table later in the text.

What VFA sensor means?

This refers to a sensor for Volatile Fatty Acids. This has also been updated in the text on line 149.

There is no methods added to create the "tree method”

In response to this we have revised the manuscript to include detailed descriptions of the methodologies used in creating the tree-based models (RF, Gradient Boosting, and DT). This includes explanations of the construction and aggregation processes in RF, the sequential tree building in Gradient Boosting, and the criteria for decision-making in DT. Additional references are now provided for a deeper understanding of these methodologies.

There is no explanation on figure 3.

A description has now been added to figure 3.

Please improve quality of figure 4.

This figure has now been updated using an image exported at higher quality.

CONCLUSION

The conclusion is too general and shallow in the specific area.

The conclusion has been updated in an atempt to make it more direct in summarising the results of the paper and highlighting the main findings. We highlight how use of ML in AD require system-specific approaches, while emphasizing the importance of comparative model performance analysis rather than literature-based model selection.

Reviewer 3 Report

Comments and Suggestions for Authors

This work deals with the use of machine learning (ML) in anaerobic digestion (AD), a systematic review. The topic is interesting, the manuscript is well organized. However, Some comments must be addressed by the author before considering the article for publication in the journal, as follows:

The abstract needs to be improved, mention practical situations (such as biogas, waste treatment).

Were the citation and impact factor parameters used in the systematic review?

Often the most recent articles are not the most important in the area studied.

The article is very comprehensive, it has topics with only one citation, for a review article, more authors should be considered.

Author Response

We appreciate the suggestions you have provided and have made an effort to change the manuscript to resolve the queries listed. Below lists our responses to your feedback:

The abstract needs to be improved, mention practical situations (such as biogas, waste treatment)

In response to this suggestion, we have revised the abstract to better reflect the practical implications, explicitly mentioning how the findings can be applied in the areas of biogas production and waste treatment processes. This addition aims to provide a clearer understanding of the real-world applications of the study.

Were the citation and impact factor parameters used in the systematic review?

Thank you for raising this point. Initially, we considered using impact factor and citation score as criteria for screening papers. However, due to the limited number of papers identified in the screening process and our confidence in the quality of Elsevier journals, we decided not to further screen papers based on these criteria.

Often the most recent articles are not the most important in the area studied.

This point has been taken into consideration. In agreement with your feedback, we have included more seminal references in our revised manuscript to support key points in the discussion, alongside recent studies. These additions aim to strengthen the arguments presented and ensure a comprehensive understanding of the subject.

Round 2

Reviewer 1 Report

Comments and Suggestions for Authors

1. The cited references are still not enough to a review. More references needs to be added.

2. There are some spelling mistakes.

Author Response

Thank you for taking the time to review this manuscript. I have incorporated your feedback into the final version. The suggestions provided in this revision process have been very valuable to improving this work.

Kind regards

Reviewer 2 Report

Comments and Suggestions for Authors

Authors have made the required modifications

Author Response

Thank you for taking the time to review this manuscript. The suggestions provided in this revision process have been very valuable to improving this work.

Kind regards

Reviewer 3 Report

Comments and Suggestions for Authors

 The authors made some corrections, in my opinion the article can be published in its present form.

Author Response

(The authors gave the same response as above.)
